# *NeuroPath*: A Neural Pathway Transformer for Joining the Dots of Human Connectomes

**Ziquan Wei**
University of North Carolina at Chapel Hill
Chapel Hill, NC 27599
`ziquanw@eamil.unc.edu`

**Tingting Dan**
University of North Carolina at Chapel Hill
Chapel Hill, NC 27599
`tingting_dan@med.unc.edu`

**Jiaqi Ding**
University of North Carolina at Chapel Hill
Chapel Hill, NC 27599
`jiaqid@cs.unc.edu`

**Guorong Wu**
University of North Carolina at Chapel Hill
Chapel Hill, NC 27599
`guorong_wu@med.unc.edu`

## Abstract

Although modern imaging technologies allow us to study connectivity between two distinct brain regions *in-vivo*, an in-depth understanding of how anatomical structure supports brain function and how spontaneous functional fluctuations emerge remarkable cognition is still elusive. Meanwhile, tremendous efforts have been made in the realm of machine learning to establish the nonlinear mapping between neuroimaging data and phenotypic traits. However, the absence of neuroscience insight in the current approaches poses significant challenges in understanding cognitive behavior from transient neural activities. To address this challenge, we put the spotlight on the coupling mechanism of structural connectivity (SC) and functional connectivity (FC) by formulating such network neuroscience question into an expressive graph representation learning problem for high-order topology. Specifically, we introduce the concept of *topological detour* to characterize how a ubiquitous instance of FC (direct link) is supported by neural pathways (detour) physically wired by SC, which forms a cyclic loop interacted by brain structure and function. In the cliché of machine learning, the multi-hop detour pathway underlying SC-FC coupling allows us to devise a novel multi-head self-attention mechanism within Transformer to capture multi-modal feature representation from paired graphs of SC and FC. Taken together, we propose a biological-inspired deep model, coined as *NeuroPath*, to find putative connectomic feature representations from the unprecedented amount of neuroimages, which can be plugged into various downstream applications such as task recognition and disease diagnosis. We have evaluated *NeuroPath* on large-scale public datasets including Human Connectome Project (HCP) and UK Biobank (UKB) under different experiment settings of supervised and zero-shot learning, where the state-of-the-art performance by our *NeuroPath* indicates great potential in network neuroscience.

## 1 Introduction

The seek of meaningful feature representations for graph topology has been extensively investigated [44, 28, 34], with widespread applications in reasoning path [16] and cycle basis [61] for knowledge graph, as well as neural fingerprint [17], junction tree autoencoder [29] and cellular Weisfeiler Leman (WL) testing [7] for molecular substructure encoding. Moreover, graph data in the realm of neuroscience research demands feature representation bearing additional neuroscience insight

38th Conference on Neural Information Processing Systems (NeurIPS 2024).

which is supposed to underline particular neurobiological mechanisms of interest. For example, substructure embedding [36, 31] and snapshot embedding [53, 3] have been proposed to characterize the phenotypic traits from structural connectivity (SC), which is a static graph and physically wired by neuronal fibers, and functional connectivity (FC) in the brain, which is a dynamic graph and supported by neural circuit overlaid on SC topology [2]. Nevertheless, the graph learning approach of physical neural pathways of SC coupled with FC is rare in related works [4, 13, 39].

The advancement of diffusion-weighted imaging (DWI) technology enables the *in-vivo* measurement of physical region-to-region connections through neuronal fibers (aka. SC). Multiple lines of neuroscience finding suggest that high-level cognition and behavior emerge from the remarkable SC-FC coupling mechanism, making the in-depth understanding of the interplay between SC and FC become the gateway to reverse engineering human mind [52, 5]. To that end, a plethora of computational models have been proposed over the past decade, including biophysical models [25, 58], graph harmonics [46, 38], network communications [19], multivariate statistical technique [41], and deep learning-based structure-function mapping [10, 49, 45]. However, the complex relationship between structural and functional connectivity is still elusive [14], particularly evidenced by the lack of direct physical pathways (formed by SC in Fig. 1 orange box) between two brain regions that exhibit functional co-activation (indicated by FC in Fig. 1 green box) [24, 42, 6]. Although the topology of SC does not necessarily always aligned with the counterpart of FC motivating previous related works, there is a converging consensus that each FC instance is supported by a sub-graph of SC, as shown in Fig. 1 orange box, where blue links constructing a path sub-graph represent the neural pathway that physically supports the red link.

We strike on this outstanding neuroscience question by lifting the concept of univariate coupling, which is limited to the correspondence between a direct link of SC and another direct link of FC, to a new paradigm of multivariate SC-FC coupling mechanism that models how the ubiquitous FC instance is associated with the detour pathway on SC topology. As shown in green box of Fig. 1, the red line represents the direct FC link between region #1 and #2 while the collection of blue lines indicates the corresponding detour pathway of SC. Although the whole-brain topology remains unchanged as brain function evolves, the functional co-activation over time is dynamically supported by different neural pathways of SC. Note, the direct FC link and SC detour form a cyclic loop inherited from both SC and FC topology. Such conceptualization is supported by various neuroscience findings that synchronization of neural activity between two brain regions is fundamentally supported by the underlying neural circuitry established by structural connectivities [23, 24, 21].

In the perspective of machine learning, as shown in the middle of Fig. 1, the overarching goal is to establish a mapping function between neuroimaging data (SC only, FC only, or both) to the phenotypic traits (such as cognitive tasks). Due to the complex nature of human cognition, however, current deep models encounter several significant challenges. First, it is less common to use the static SC only for the prediction of evolving cognitive states, largely due to the ill-posed setting which boils down to a one-to-many mapping [50]. Alternatively, tremendous efforts have been made to predict cognitive states based on time series of neural activity [3] and FC topology [4]. However, it has been frequently reported that massive inter-subject variations of FC often predominate the intrinsic task-relevant patterns, such variations in real-world data are demonstrated in Sec. 2.1. Thus, current learning-based approaches have difficulties in scaling up to large-size population data. Furthermore, limiting consideration to either SC or FC alone fails to capture the holistic nature of the brain as a dynamic system, resulting in existing approaches having limited power to uncover new insight of complex relationship between SC and FC [39]. In this regard, it is natural to combine the information of SC and FC in a multi-modality learning scenario [51, 30, 57]. In general, current research focuses on information fusion at node or link level. Few attention has been paid to integrating SC and FC by combining the power of machine learning and insight of cognitive neuroscience.

Taken together, we propose a novel deep model, coined *NeuroPath*, to (1) enhance the machine performance of predicting cognitive status using both SC and FC information and (2) uncover new neuroscience underpinning of SC-FC coupling mechanism. In a nutshell, we conceptualize that the effective way to advance our current understanding of cognitive neuroscience is to characterize the latent SC-FC coupling mechanism from neuroimaging data. To land this conceptualization into an end-to-end deep model, we put the spotlight on devising a new multi-head self-attention (MHSA) component by capitalizing on the multivariate SC-FC coupling mechanism. As shown in Fig. 1 grey box, nodes of multiple paths detouring an FC link on SC are transformed respectively by multiple heads of self-attention in one layer. On top of this, we learn putative feature representations from

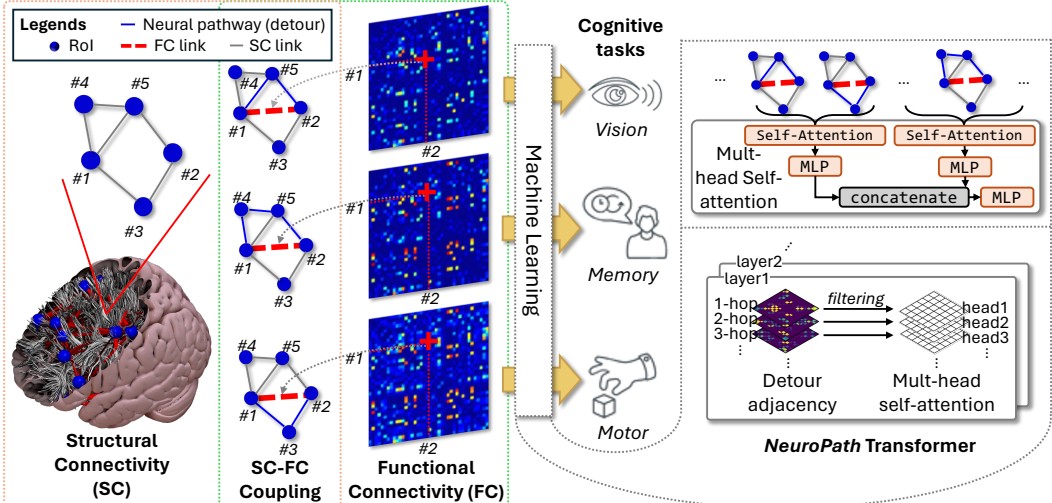

Figure 1: Planting a novel multivariate SC-FC coupling mechanism to explainable deep model. Orange box: The structural connectivity (SC) denoted by grey links represents the strength of neurological fiber that physically connects two brain regions. SC is relatively static given the neural activities are transient, e.g. cognitive tasks. Green box: The functional connectivity (FC) is commonly considered as the brain network topology [48] since SC is static for different cognitive tasks. The overlapping area of orange and green boxes: the multivariate SC-FC coupling mechanism, where a neural pathway (detour) is constructed by multiple SC links to support one FC link. Grey box: *NeuroPath* Transformer using a new MHSA module filtered by adjacency matrices emits the representation of multi-hop detours.

the cyclic loop capturing complete information on how the underlying functional fluctuations are supported by the neural pathway (aka. SC detour) with different lengths.

Our major contribution has three folds. *First*, we present a novel multivariate SC-FC coupling mechanism that allows us to develop an explainable deep model with great neuroscience insight and guarantee from graph theory. *Second*, we present a Transformer-based deep model that is scalable to utilize existing large population of neuroimaging data. *Third*, we have validated our *NeuroPath* on large-scale public datasets with a total of 10,886 fMRI scans including Human Connectome Project (HCP), UK Biobank (UKB), Alzheimer's Disease Neuroimaging Initiative (ADNI), and Open Access Series of Imaging Studies (OASIS). We have evaluated (1) the accuracy for predicting phenotypic traits (cognitive status in healthy brains and disease risk for aging brains) (2) model interpretability in identifying latent neural pathways, and (3) the clinical value of our model in terms of robustness and zero-shot learning, where promising results suggests potential application in computational neuroscience field.

## 2 Preliminaries

### 2.1 Motivations

Massive inter-subject variations of FC often predominate the intrinsic task-relevant patterns. Take the HCP data [9] as an example in which the data heterogeneity issue has been harmonized by using standardized imaging protocol. As shown in Fig. 2, we first identify a set of brain regions which exhibit significant difference between resting stage and VISMOTOR tasking [9] in a paired *t*-test using detour degree and FC degree[1], respectively, where the identified regions are colored based on *p*-values. Next, we examine the location of these regions in each functional sub-networks (manually defined using neuroscience domain knowledge). For comparison, we display the overlap ratio (y-axis) between the number identified regions and the total number of regions in each sub-network (x-axis) at the bottom of Fig. 2. Current neuroscience findings suggest that resting state is relevant to default mode network (red circle) while VISMOTOR task is associated with visual (orange circle) and sensorimotor areas (green circle). In this regard, the detour degree achieves

---

[1]Detailed steps of paired *t*-tests can be found in Appendix.

overall higher discrimination power than FC to the extent that most of identified regions are aligned with neuroscience finding with much less false positives in other non-relevant regions. The same findings are observed on diverse populations by gender in Appendix Fig. 8. The evidence shown in Fig. 2 underscores the importance of including SC detour in finding putative functional biomarkers, prompting us to further integrate the SC-FC coupling mechanism into the design of *NeuroPath*.

## 2.2 Relevant Machine Learning Work for Brain Connectomes

Tremendous efforts have been made to perform graph learning on human connectome data. The most representative work includes graph convolutional network (GCN [33, 36]) and transformer-based deep models [3, 31]. Recently, most of popular GNN models and their application in neuroscience have been reviewed in [4].

Inspired by the great success of Transform [56] in NLP and CV, various graph transformer models have been proposed in graph learning. For example, GraphGPS [47] and TokenGT [32] models use either a structure encoding or token of graph elements as the initial graph feature representations. Furthermore, a self-attention gating mechanism is proposed in [26] using an additional branch of edge embedding to implement the augmentation of self-attention. By integrating original node features and structural embeddings into an augmented self-attention module, Graphormer [62] shows the same expressibility as conventional graph neural networks (GNN), which has been proved in [63] that such augmented self-attention is strictly more expressive than low-order WL for distinguishing non-isomorphic graphs [43]. In the following, we present our *NeuroPath* where the novel biological-inspired self-attention mechanism yields more expressive power of graph representation than Graphormer [62].

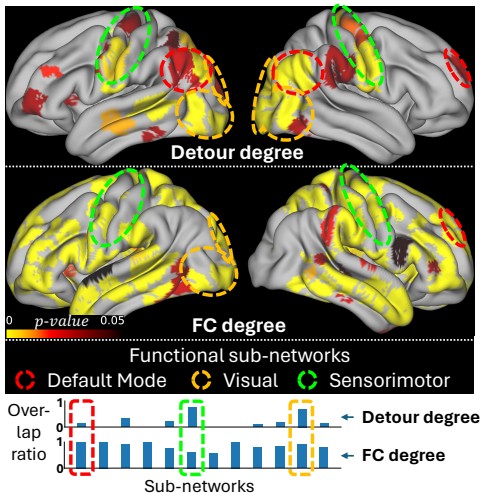

Figure 2: Motivation of integrating SC-FC coupling in neural networks. Top: Brain regions manifesting significant resting state vs VISMOTOR difference using detour degree and FC degree. Color indicate the $p$-values ($0 \leq p \leq 0.05$) in paired $t$-test. Bottom: The overlap ratio (y-axis) between the number of identified significant regions and the total number of brain regions in each pre-defined functional sub-networks (x-axis). Specifically, we examine the identified regions in default mode (red circle/box), visual (orange circle/box), and sensorimotor (green circle/box) networks since they are closely associated with resting stage vs VISMOTOR difference.

# 3 *NeuroPath* – A Biological-Inspired Transformer for Human Connectomes

**Problem formulation** Given that the detour degree can profile nodes highly associated with functional sub-networks, the problem in this work is formulated as how to plant this neuroscience insight of topological detour from SC-FC coupling into deep neural networks.

**Notations.** Assume SC and FC are denoted by adjacency matrices $\mathbf{A^S} \in \mathbb{R}^{N \times N}$ and $\mathbf{A^F} \in \mathbb{R}^{N \times N}$, respectively, where $A_{ij}^{(\cdot)}$ is the connectivity between $i^{th}$ and $j^{th}$ region ($i, j = 1, ..., N$). In practice, SC and FC are calculated by the subject-wise normalized water matter (WM) fiber counts and the Pearson's correlation coefficient of neural signals, respectively. Furthermore, we use $\hat{\mathbf{A}}^{(\cdot)}$ to denote the binary adjacency matrix after high-pass filtering and adding self-loop for either SC or FC. Node attribute is denoted by $\mathbf{X} \in \mathbb{R}^{N \times C}$, where $C$ is the feature dimension.

**Definition of detour in the context of SC-FC coupling.** To capture multi-scale topology, we employ random walk on graph of SC, yielding a set of multi-hop detour adjacency matrices as follows:

**Definition 3.1** (Detour adjacency matrix). $\mathbf{D}^h$, a binary matrix shapes of $N \times N$ and stores if a link of FC is associated with a $h$-hop topological detour, where a 1-hop detour is equivalent to an edge. It is obtained by element-wise production between binary matrices $\mathbf{D}^h := \left( (\hat{\mathbf{A}}^{\mathbf{S}})^h > 0 \right) \cdot \left( \hat{\mathbf{A}}^{\mathbf{F}} \right)$, where $1 \leq h \leq H$, and $H$ is the maximum length of detour.

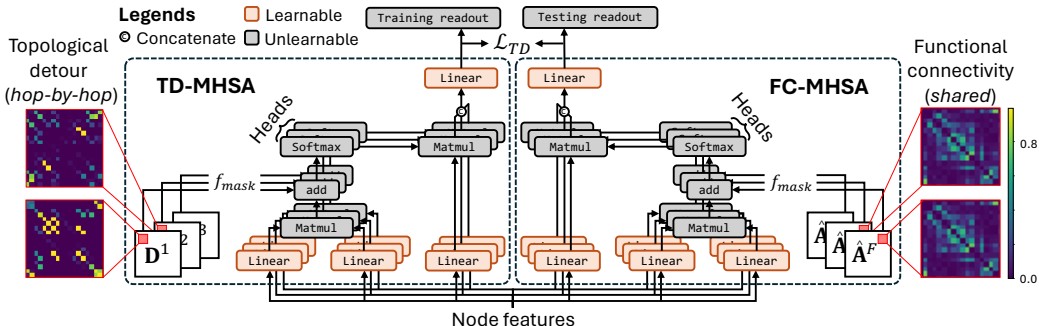

Figure 3: Framework of twin branch, topological detour filtered multi-head self-attention (TD-MHSA) and functional connectivity filtered multi-head self-attention (FC-MHSA), for node feature transformation in *NeuroPath*, where training/testing readout indicates different branch is used for training/testing stage.

Note that $\mathbf{D}^h$ avoids finding all simple paths by our model, it significantly reduces computational costs with sufficient power for the neural pathways representation as discussed in Sec. 3.2.

## 3.1 Network Architecture of *NeuroPath*

*NeuroPath* is designed with twin branch as shown in Figure 3, where two branches in the model have identical frameworks of the multi-head self-attention (MHSA) from the Transformer encoder [56]. This twin branch is designed for feature representation to fuse SC and FC information by being consistent between FC link representation and SC detour representation. Given a set of learnable parameters $\bar{\mathbf{W}}, \hat{\mathbf{W}} \in \mathbb{R}^{(HC) \times C}$ and $\bar{\boldsymbol{\alpha}}_h, \bar{\boldsymbol{\beta}}_h, \bar{\boldsymbol{\gamma}}_h, \hat{\boldsymbol{\alpha}}_h, \hat{\boldsymbol{\beta}}_h, \hat{\boldsymbol{\gamma}}_h \in \mathbb{R}^{C \times C}$ where $h = 1, \ldots, H$, $H$ is head number of MHSA, and $C$ denotes feature dimension, the branch TD-MHSA is then defined as

$$f_{TD}(\mathbf{X}) = \text{Concat}\left(\bar{\mathbf{X}}_1, \ldots, \bar{\mathbf{X}}_H\right) \bar{\mathbf{W}}, \tag{1}$$

where $\bar{\mathbf{X}}_h = \text{Softmax}\left((\mathbf{X}\bar{\boldsymbol{\alpha}}_h)(\mathbf{X}\bar{\boldsymbol{\beta}}_h)^T + f_{mask}(\mathbf{D}^h)\right)(\mathbf{X}\bar{\boldsymbol{\gamma}}_h)$. Similarly, the FC-MHSA branch is defined as

$$f_{FC}(\mathbf{X}) = \text{Concat}\left(\hat{\mathbf{X}}_1, \ldots, \hat{\mathbf{X}}_H\right) \hat{\mathbf{W}}, \tag{2}$$

where $\hat{\mathbf{X}}_h = \text{Softmax}\left((\mathbf{X}\hat{\boldsymbol{\alpha}}_h)(\mathbf{X}\hat{\boldsymbol{\beta}}_h)^T + f_{mask}(\hat{\mathbf{A}}^{\mathbf{F}})\right)(\mathbf{X}\hat{\boldsymbol{\gamma}}_h)$. Adding masks to attention maps is a mature approach for only involving interested nodes in the self-attention mechanism. To achieve this, $f_{mask}$ here is defined as filling negative infinities to false slots and zeros to true slots of the binary adjacency matrix so that Softmax can ignore false slots in both branches.

Loss function during training is set as the downstream application objective along with the consistency constraint loss $\mathcal{L}_{TD}$, which is defined as

$$\mathcal{L}_{TD} = ||f_{TD}(\mathbf{X}) - f_{FC}(\mathbf{X})||^2. \tag{3}$$

Taking classification as an example, the final loss $\mathcal{L} = \text{CELoss}\left(label, \mathbf{Y}\right) + \mathcal{L}_{TD}$, where the logits $\mathbf{Y} = \boldsymbol{\rho}^{-\frac{1}{2}}\hat{\mathbf{A}}^{\mathbf{F}}\boldsymbol{\rho}^{-\frac{1}{2}}f_{TD}(\mathbf{X})\boldsymbol{\Theta}$ with learnable parameters $\boldsymbol{\Theta} \in \mathbb{R}^{C \times n}$, $n$ is class number and $\boldsymbol{\rho}$ is degree of brain network adjacency $\hat{\mathbf{A}}^{\mathbf{F}}$.

## 3.2 Neural Pathway Representation by *NeuroPath*

TD-MHSA strictly follows the pathway of topological detour after SC-FC coupling to produce node features hop-by-hop corresponding to each head of self-attention. In fact, the $i$-th node feature representation, $f_{TD}(\mathbf{X})_i$, is the weighted sum of nodes among half of the neural pathways in the range of $H$-hop after expanding Eq. 1. Detailed proof can be found in Appendix.

**Fact 3.1.** *The top pathway representations are obtained by* $\arg\max_{j,h}\left(\frac{1}{h}\sum_{j \in \mathbf{p}} \mathbf{S}_{ij}\bar{\boldsymbol{\gamma}}_h\bar{\mathbf{W}}_{i \sim j}\right)$, *where* $\mathbf{S}$ *denotes the softmax of self-attention,* $\mathbf{p} \subset \mathbf{P}_i^H$ *is a set of node index of a path and* $\mathbf{P}_i^H$ *is the node collection of neural pathways within $H$-hop starting at $i$-th node.*

Fact 3.1 indicates the power of our *NeuroPath* for high-order graph substructure. PathNN [40] models all simple paths has $O(N^H)$ computational complexity by sorting all paths in advance. In contrast,

*NeuroPath* can model important paths using only $O(N^2)$ computational complexity and does not require any pre-computation of high-order topology.

Regarding the neuroscience knowledge that FC is relatively denser than SC with the evidence of significant indirect SC links [39], FC adjacency is also denser than the TD adjacency according to the Def. 3.1. Meanwhile, the consistency constraint $\mathcal{L}_{TD}$ cooperates with the twin branch so that FC-MHSA produces the expressive node feature representation as well. Consequently, existing states of FC-MHSA can hold the same power of TD-MHSA by a consistent feature representation, and hence benefit the prediction by this SC-FC coupling.

## 4  Experimental Results

The experiments are designed for the following four Research Questions (RQ). **RQ1:** How is performance on neural activity recognition and cognition disordering disease diagnosis compared to state-of-the-art (SOTA) brain models and graph transformers? **RQ2:** Can the brain network representation by *NeuroPath* being consistent on zero-shot learning experiments? **RQ3:** Does performance enhancement by *NeuroPath* align with intuitions by ablations of model framework and neural pathway length? **RQ4:** What is the pattern of neural pathways contributing to prediction? To thoroughly answer those questions, four public datasets are used in our experiments. Detailed data preprocessing and profiles can be found in Appendix.

**The Lifespan Human Connectome Project Aging (HCPA)** dataset [9] is instrumental in task recognition research, offering a comprehensive view of the aging process. It includes data from 717 subjects, encompassing both fMRI ($n$=4,863) and DWI ($n$=716). It includes data from three brain tasks associated with memory and sensory-motor: VISMOTOR, CARIT, and FACENAME, and the resting state. In the related experiments, these tasks are treated as (1) a four-class classification problem and (2) resting/tasking classification for zero-shot learning. We partition brain regions using AAL atlas [55] in experiments except for ablation studies using Gordon atlas [20].

**United Kingdoms Biobank (UKB)** dataset is a large-scale dataset with MRI data. There are fMRI ($n$=5,483) and DWI ($n$=3,162) preprocessed by the same algorithm as HCPA. It includes data from one brain task that engages cognitive and sensory-motor [22, 37]. Data is treated as a two-class classification in the following experiments. Brain regions are partitioned by Gordon atlas [20].

**Alzheimer's Disease Neuroimaging Initiative (ADNI)** dataset [59] serves as an invaluable resource, featuring a collection of pre-processed fMRI ($n$=138) and DWI ($n$=135) with AAL parcellation in our experiment. Additionally, ADNI includes clinical diagnostic labels, encompassing a spectrum of cognitive states: Cognitive Normal (CN), Subjective Memory Complaints (SMC), Early-Stage Mild Cognitive Impairment (EMCI), Late-Stage Mild Cognitive Impairment (LMCI), and Alzheimer's Disease (AD). Considering the data balance issue, we simplified these categories into two broad groups based on disease severity: we combined CN, SMC, and EMCI into 'CN' group, while LMCI and AD were grouped as the 'AD' group.

**Open Access Series of Imaging Studies (OASIS)** dataset [35] presents a substantial collection of data from 924 subjects, comprising 3,322 fMRI sessions in total. Among the dataset, fMRI ($n$=402) and DWI ($n$=362) are pre-processed with Destrieux parcellation [15]. Our experiment focused on binary classification: subjects in preclinical stages of AD or those manifesting dementia-related conditions are under the 'AD' group, while healthy individuals are under the 'CN' group.

All datasets are split as train/validation in a 5-fold cross-validation according to the subject index to prevent data leakage. Node attributes of brain network setting as Blood-Oxygen Level Dependent (BOLD) signal or correlation (CORR) between BOLD signals are performing vary on different datasets [48]. Furthermore, the deep model performance is also affected by using a short-length or full-length BOLD which is considered a dynamic or static brain network, respectively [48]. To evaluate *NeuroPath* under all situations, four combinations of CORR/BOLD and static/dynamic are all tested in our experiments.

Competitive methods, two baselines (MLP and GCN), three SOTA brain models (BrainGNN [36], BNT [31], and BolT [3]), and two SOTA general graph transformers (Graphormer [62] and NAG-phormer [12]) are chosen for comparison with *NeuroPath*, where MLP and GCN both have one layer of the vanilla framework followed by batch normalization and activation for feature representation

Table 1: Performance comparison on HCPA and UKB datasets. Colored numbers indicate ranking in the first, second, and third place.

| | HCPA CORR | | HCPA BOLD | | UKB CORR | | UKB BOLD | |
|---|---|---|---|---|---|---|---|---|
| | static | dynamic | static | dynamic | static | dynamic | static | dynamic |
| ● Accuracy | | | | | | | | |
| MLP | $96.01_{\pm0.50}$ | $92.57_{\pm0.32}$ | $93.42_{\pm0.58}$ | $83.64_{\pm1.33}$ | $99.00_{\pm0.15}$ | $97.70_{\pm0.32}$ | $99.05_{\pm0.48}$ | $96.42_{\pm0.60}$ |
| GCN | $95.85_{\pm0.93}$ | $91.95_{\pm0.45}$ | $92.94_{\pm0.58}$ | $84.60_{\pm0.45}$ | $99.00_{\pm0.22}$ | $97.54_{\pm0.24}$ | $99.31_{\pm0.33}$ | $93.39_{\pm0.71}$ |
| BrainGNN | $90.85_{\pm1.35}$ | $86.06_{\pm2.64}$ | $89.38_{\pm2.88}$ | $72.62_{\pm3.33}$ | $97.54_{\pm0.52}$ | $95.32_{\pm1.68}$ | $90.33_{\pm2.72}$ | $86.11_{\pm4.04}$ |
| BNT | $97.92_{\pm0.65}$ | $94.18_{\pm0.35}$ | $92.57_{\pm1.19}$ | $86.55_{\pm0.37}$ | $98.71_{\pm0.34}$ | $97.15_{\pm0.49}$ | $98.64_{\pm0.18}$ | $95.98_{\pm0.44}$ |
| BolT | $96.40_{\pm0.41}$ | $91.68_{\pm0.38}$ | $95.78_{\pm0.55}$ | $91.92_{\pm0.69}$ | $99.13_{\pm0.33}$ | $97.61_{\pm0.23}$ | $99.29_{\pm0.26}$ | $98.22_{\pm0.31}$ |
| Graphormer | $78.80_{\pm5.89}$ | $78.73_{\pm1.91}$ | $59.63_{\pm6.07}$ | $65.01_{\pm3.84}$ | $92.76_{\pm10.05}$ | $81.98_{\pm9.83}$ | $86.82_{\pm12.42}$ | $55.56_{\pm21.08}$ |
| NAGphormer | $93.67_{\pm0.96}$ | $90.73_{\pm0.64}$ | $94.76_{\pm1.15}$ | $82.02_{\pm1.77}$ | $98.79_{\pm0.35}$ | $96.83_{\pm0.36}$ | $99.22_{\pm0.36}$ | $92.90_{\pm0.69}$ |
| *NeuroPath* | $96.69_{\pm0.54}$ | $92.76_{\pm0.52}$ | $95.03_{\pm1.93}$ | $87.54_{\pm0.77}$ | $99.22_{\pm0.24}$ | $97.77_{\pm0.21}$ | $99.59_{\pm0.21}$ | $94.12_{\pm0.75}$ |
| ● F1 score | | | | | | | | |
| MLP | $96.01_{\pm0.49}$ | $92.52_{\pm0.35}$ | $93.42_{\pm0.58}$ | $82.86_{\pm1.68}$ | $99.00_{\pm0.15}$ | $97.69_{\pm0.32}$ | $99.05_{\pm0.49}$ | $96.42_{\pm0.60}$ |
| GCN | $95.85_{\pm0.95}$ | $91.90_{\pm0.41}$ | $92.98_{\pm0.60}$ | $83.95_{\pm0.37}$ | $99.00_{\pm0.22}$ | $97.53_{\pm0.24}$ | $99.31_{\pm0.33}$ | $93.36_{\pm0.71}$ |
| BrainGNN | $90.92_{\pm1.41}$ | $85.43_{\pm3.37}$ | $89.38_{\pm2.92}$ | $64.40_{\pm6.67}$ | $97.54_{\pm0.52}$ | $95.30_{\pm1.71}$ | $90.35_{\pm2.70}$ | $86.09_{\pm4.18}$ |
| BNT | $97.92_{\pm0.66}$ | $94.16_{\pm0.35}$ | $92.57_{\pm1.22}$ | $86.45_{\pm0.40}$ | $98.71_{\pm0.34}$ | $97.15_{\pm0.49}$ | $98.64_{\pm0.18}$ | $95.97_{\pm0.43}$ |
| BolT | $96.38_{\pm0.43}$ | $91.66_{\pm0.39}$ | $95.78_{\pm0.57}$ | $91.76_{\pm0.78}$ | $99.13_{\pm0.34}$ | $97.60_{\pm0.23}$ | $99.29_{\pm0.26}$ | $98.22_{\pm0.31}$ |
| Graphormer | $77.29_{\pm7.15}$ | $75.26_{\pm2.66}$ | $53.05_{\pm5.81}$ | $57.04_{\pm1.46}$ | $92.67_{\pm10.25}$ | $80.19_{\pm11.35}$ | $86.54_{\pm13.03}$ | $50.12_{\pm27.58}$ |
| NAGphormer | $93.69_{\pm0.95}$ | $90.64_{\pm0.68}$ | $94.76_{\pm1.16}$ | $81.06_{\pm2.03}$ | $98.79_{\pm0.35}$ | $96.82_{\pm0.35}$ | $99.22_{\pm0.36}$ | $92.88_{\pm0.68}$ |
| *NeuroPath* | $96.70_{\pm0.54}$ | $92.72_{\pm0.54}$ | $95.09_{\pm1.86}$ | $87.03_{\pm0.95}$ | $99.22_{\pm0.24}$ | $97.77_{\pm0.21}$ | $99.59_{\pm0.21}$ | $94.11_{\pm0.75}$ |

Table 2: Performance comparison on ADNI and OASIS datasets. Colored numbers indicate ranking in the first, second, and third place.

| | ADNI CORR | | ADNI BOLD | | OASIS CORR | | OASIS BOLD | |
|---|---|---|---|---|---|---|---|---|
| | static | dynamic | static | dynamic | static | dynamic | static | dynamic |
| ● Accuracy | | | | | | | | |
| MLP | $79.26_{\pm10.34}$ | $82.68_{\pm5.71}$ | $80.67_{\pm7.26}$ | $82.93_{\pm6.35}$ | $89.28_{\pm3.58}$ | $89.32_{\pm3.18}$ | $88.99_{\pm3.52}$ | $89.02_{\pm3.25}$ |
| GCN | $84.22_{\pm6.92}$ | $83.30_{\pm6.30}$ | $80.67_{\pm7.26}$ | $83.53_{\pm5.39}$ | $88.80_{\pm2.88}$ | $88.30_{\pm3.54}$ | $88.27_{\pm4.87}$ | $88.49_{\pm3.16}$ |
| BrainGNN | $82.07_{\pm6.86}$ | $83.30_{\pm5.42}$ | $82.07_{\pm6.86}$ | $83.42_{\pm6.05}$ | $89.29_{\pm4.75}$ | $89.65_{\pm3.31}$ | $87.76_{\pm4.64}$ | $89.27_{\pm3.36}$ |
| BNT | $82.81_{\pm6.47}$ | $83.30_{\pm6.30}$ | $82.67_{\pm4.40}$ | $84.33_{\pm6.99}$ | $89.02_{\pm3.48}$ | $89.98_{\pm2.75}$ | $88.75_{\pm4.36}$ | $89.57_{\pm3.02}$ |
| BolT | $82.00_{\pm3.51}$ | $80.34_{\pm2.82}$ | $81.41_{\pm7.08}$ | $80.80_{\pm7.53}$ | $88.30_{\pm3.77}$ | $88.97_{\pm3.04}$ | $87.54_{\pm4.62}$ | $88.50_{\pm3.35}$ |
| Graphormer | $82.74_{\pm5.89}$ | $83.28_{\pm5.80}$ | $83.48_{\pm5.31}$ | $81.28_{\pm6.58}$ | $88.55_{\pm4.22}$ | $88.57_{\pm3.18}$ | $87.49_{\pm5.19}$ | $88.98_{\pm3.26}$ |
| NAGphormer | $82.74_{\pm5.89}$ | $82.79_{\pm5.82}$ | $81.33_{\pm6.09}$ | $82.17_{\pm5.73}$ | $89.53_{\pm3.33}$ | $88.64_{\pm3.85}$ | $89.02_{\pm3.48}$ | $89.21_{\pm3.44}$ |
| *NeuroPath* | $85.56_{\pm4.97}$ | $83.82_{\pm3.94}$ | $83.48_{\pm5.31}$ | $83.68_{\pm5.64}$ | $90.01_{\pm3.42}$ | $89.49_{\pm3.33}$ | $89.02_{\pm3.48}$ | $89.21_{\pm3.44}$ |
| ● F1 score | | | | | | | | |
| MLP | $74.72_{\pm8.67}$ | $77.98_{\pm5.79}$ | $74.96_{\pm9.17}$ | $76.75_{\pm7.08}$ | $87.05_{\pm5.00}$ | $86.74_{\pm4.81}$ | $85.27_{\pm4.82}$ | $85.02_{\pm5.03}$ |
| GCN | $78.53_{\pm9.76}$ | $76.95_{\pm8.17}$ | $76.19_{\pm8.50}$ | $77.87_{\pm5.66}$ | $84.75_{\pm5.56}$ | $85.71_{\pm4.10}$ | $85.56_{\pm5.55}$ | $85.86_{\pm3.97}$ |
| BrainGNN | $76.57_{\pm10.01}$ | $79.14_{\pm8.02}$ | $75.11_{\pm9.69}$ | $78.82_{\pm6.96}$ | $86.07_{\pm5.71}$ | $85.12_{\pm4.90}$ | $84.94_{\pm5.22}$ | $84.50_{\pm5.00}$ |
| BNT | $79.68_{\pm6.15}$ | $78.71_{\pm6.67}$ | $80.16_{\pm8.01}$ | $80.50_{\pm8.40}$ | $86.07_{\pm3.19}$ | $86.73_{\pm3.57}$ | $85.32_{\pm4.85}$ | $85.67_{\pm4.04}$ |
| BolT | $79.64_{\pm4.33}$ | $76.89_{\pm7.75}$ | $76.68_{\pm8.77}$ | $77.92_{\pm8.62}$ | $85.49_{\pm3.85}$ | $84.91_{\pm4.76}$ | $84.91_{\pm4.76}$ | $84.70_{\pm4.39}$ |
| Graphormer | $78.14_{\pm6.03}$ | $78.29_{\pm5.22}$ | $77.78_{\pm5.51}$ | $76.74_{\pm6.95}$ | $84.77_{\pm5.24}$ | $85.67_{\pm3.72}$ | $85.44_{\pm4.73}$ | $84.40_{\pm4.77}$ |
| NAGphormer | $76.57_{\pm6.67}$ | $76.46_{\pm6.93}$ | $75.40_{\pm8.58}$ | $77.80_{\pm7.01}$ | $85.61_{\pm4.79}$ | $84.76_{\pm4.58}$ | $83.87_{\pm5.02}$ | $84.48_{\pm4.58}$ |
| *NeuroPath* | $83.29_{\pm4.45}$ | $79.93_{\pm5.83}$ | $77.35_{\pm7.35}$ | $78.05_{\pm5.88}$ | $86.37_{\pm5.03}$ | $86.26_{\pm4.48}$ | $87.02_{\pm3.77}$ | $85.01_{\pm4.48}$ |

Table 3: Conclusion of performance: Average rank of methods on different scenarios and evaluation metrics. '**Bold**' and 'underline' denote the first and the second place, respectively.

| | MLP | GCN | BrainGNN | BNT | BolT | Graphormer | NAGphormer | *NeuroPath* |
|---|---|---|---|---|---|---|---|---|
| HCPA | 4.0 | 4.5 | 7.0 | 2.8 | 2.5 | 8.0 | 5.3 | **2.0** |
| UKB | 3.0 | 3.75 | 7.0 | 5.0 | 2.3 | 8.0 | 5.3 | **1.8** |
| ADNI | 6.9 | 4.4 | 4.1 | 2.1 | 5.8 | 4.6 | 5.8 | **1.6** |
| OASIS | 3.3 | 5.3 | 4.4 | 2.8 | 6.5 | 6.4 | 5.0 | **2.5** |

and one layer of graph convolution for graph prediction. All train/validate settings such as random seed and learning rate are set as the same.

## 4.1 Performance of Neural Activity Classification

Performance on datasets HCPA and UKB is listed in Table 1. Except for UKB BOLD dynamic, *NeuroPath* can rank in the top two places under all data settings on the accuracy and the weighted F1 score. The highest accuracy of neural activity classification on all settings of UKB can be achieved by *NeuroPath* as 99.59%, while the best by other models, 99.31%, is achieved by the GCN rather than other fancy models implying they are rarely capturing features from neuroscience senses. Regarding HCPA dataset, SOTA brain models have more than double the number of parameters than our model as listed in Appendix. Nonetheless, we can hold the second place for all data settings in HCPA since *NeuroPath* learn from the physical neural pathways, while the general graph transformers can be even worse than the vanilla MLP and GCN since they do not involve neuroscience knowledge.

## 4.2 Performance of Cognition Disordering Diagnosis

The performance of AD/CN classification on datasets ADNI and OASIS is listed in Table 2. *NeuroPath* has the top three performance among all datasets with four different data settings except for the F1 score on dynamic OASIS BOLD. The neuroscience insight of SC-FC coupling benefits *NeuroPath* achieved the best accuracy on $5/8$ cases and the best F1 score on $3/8$ cases. It is worth noting that *NeuroPath* is the only one that has over 90% accuracy in these experiments. Class unbalance in ADNI and OASIS datasets makes the F1 score lower than accuracy. Nevertheless, *NeuroPath* has top scores on both two datasets, 83.29% and 87.02%, respectively, under all data settings. Given the challenges of more neural pathways present in dynamic graphs that have a higher degree as listed in Appendix, our performance slightly drops to against others. In summary, *NeuroPath* shows superior performance than existing SOTA models on static graphs of various data with the benefit of our novel multivariate SC-FC coupling.

## 4.3 Zero-shot Learning

Despite neural activity classification and cognitive disordering diagnosis are two types of experiments usually tested in computational neuroscience research, zero-shot learning by training and validating the model on one dataset and testing on another dataset is rarely present in the field. Since for both resting/tasking and AD/CN classification, we all have two datasets, it is feasible to test our explainable deep model by zero-shot learning to show the clinical value. The experimental setting is consistent as above with a 5-fold cross-validation to choose the best model state on the train-val dataset, and then test it on validating a set of the corresponding fold of another dataset.

Table 4: Zero-shot learning between four datasets using BOLD as node attributes under different data settings, where resting/tasking state classification is tested for HCPA and UKB datasets, F1 scores are listed, and '**bold**' and 'underline' denote the first and the second rank, respectively.

|  | OASIS→ADNI | | ADNI→OASIS | |
|---|---|---|---|---|
|  | static | dynamic | static | dynamic |
| Graphormer | $77.63_{\pm 2.89}$ | $77.24_{\pm 7.34}$ | $79.69_{\pm 7.71}$ | $\mathbf{83.55}_{\pm 6.90}$ |
| NAGphormer | $73.11_{\pm 5.90}$ | $78.09_{\pm 7.24}$ | $69.63_{\pm 10.99}$ | $78.09_{\pm 7.08}$ |
| *NeuroPath* | $\mathbf{79.78}_{\pm 3.53}$ | $\mathbf{81.57}_{\pm 7.24}$ | $\mathbf{80.03}_{\pm 8.50}$ | $79.65_{\pm 6.35}$ |

|  | HCPA→UKB | | UKB→HCPA | |
|---|---|---|---|---|
|  | static | dynamic | static | dynamic |
| Graphormer | $39.09_{\pm 28.14}$ | $50.97_{\pm 4.01}$ | $57.78_{\pm 14.50}$ | $64.36_{\pm 7.90}$ |
| NAGphormer | $74.49_{\pm 4.01}$ | $70.17_{\pm 1.31}$ | $89.77_{\pm 0.94}$ | $73.44_{\pm 0.70}$ |
| *NeuroPath* | $\mathbf{91.29}_{\pm 2.10}$ | $\mathbf{72.08}_{\pm 2.15}$ | $\mathbf{90.61}_{\pm 3.65}$ | $\mathbf{75.62}_{\pm 2.98}$ |

Given that the three SOTA brain models are all sensitive to the node number of the graph, they are not compatible with zero-shot learning. Therefore, F1 scores are listed in Table 4 in comparison against two general graph transformers, which are designed for various vertex-cardinality. It is worth highlighting that our model outperforms all competitors under different data settings except for ADNI→OASIS dynamic. Specifically, *NeuroPath* can surpass the best of others with a 16.8 score for HCPA→UKB static. Although the performance of zero-shot learning by *NeuroPath* still has an observable gap to the fully supervised version listed in Table 1 and 2, results here show that the neural pathway pattern learned by *NeuroPath* is more consistent than FC pattern across datasets.

## 4.4 Ablation of Hyperparameter $H$

Since the length of the neural pathway is limited by the head number of two branches in *NeuroPath*, the ablation of the head number in MHSA leads to various performances of our model. As shown in Fig. 4, we set head number $H$ from 1 to 8 keeping all other settings remain the same. The green

Table 5: Ablation of none branch (a vanilla Transformer), single branch, and twin branch of *NeuroPath* on four datasets with static BOLD as node attributes, where '**bold**' and 'underline' denote the first and the second rank, respectively.

| | ADNI | | OASIS | | HCPA | | UKB | |
|---|---|---|---|---|---|---|---|---|
| | Accuracy | F1 score | Accuracy | F1 score | Accuracy | F1 score | Accuracy | F1 score |
| None | $82.42_{\pm5.98}$ | $78.65_{\pm7.37}$ | $88.52_{\pm3.48}$ | $86.19_{\pm3.81}$ | $97.53_{\pm0.50}$ | $97.53_{\pm0.51}$ | $99.53_{\pm0.22}$ | $99.53_{\pm0.22}$ |
| w/ TD-MHSA | $82.74_{\pm7.88}$ | $77.51_{\pm9.39}$ | $89.05_{\pm3.99}$ | $86.11_{\pm4.32}$ | $97.33_{\pm0.44}$ | $97.34_{\pm0.43}$ | $99.10_{\pm0.13}$ | $99.10_{\pm0.13}$ |
| w/ FC-MHSA | $81.93_{\pm3.25}$ | $80.97_{\pm4.20}$ | $89.31_{\pm4.36}$ | $\mathbf{86.58}_{\pm5.87}$ | $97.72_{\pm0.34}$ | $97.72_{\pm0.34}$ | $99.25_{\pm0.18}$ | $99.25_{\pm0.18}$ |
| w/ both | $\mathbf{85.56}_{\pm4.97}$ | $\mathbf{83.29}_{\pm4.45}$ | $\mathbf{90.01}_{\pm3.42}$ | $86.37_{\pm5.03}$ | $\mathbf{98.23}_{\pm0.45}$ | $\mathbf{98.23}_{\pm0.45}$ | $\mathbf{99.59}_{\pm0.21}$ | $\mathbf{99.59}_{\pm0.21}$ |

curves shown in Fig. 4 demonstrate clear peaks at $H = 7$ on datasets HCPA and UKB that are processed with a finer parcellation of 333 regions. In comparison, for the atlas used on datasets ADNI and OASIS has no more than 200 regions, the peak disappeared with a flat trend of F1 score suggesting longer pathways are less contributing to the prediction under fewer regions. This is aligned with the nature of more hops needed to construct the same neural pathway where regions are more but smaller on HCPA and UKB than ADNI and OASIS.

## 4.5 Ablation of Model Architecture

Twin branch is a core design of *NeuroPath* to plant the insight of SC-FC coupling to deep model. The ablation of using none or a single branch can illustrate the effectiveness of our design. As listed in Table 5, *NeuroPath* with twin branch has the best performance on all datasets except for the F1 score on OASIS. Despite TD-MHSA using the detour adjacency that can detect brain communities as mentioned in Sec. 2.1, SC-FC fused feature representation by using the consistency constraint $\mathcal{L}_{TD}$ for twin branch has more contribution to the performance.

## 4.6 Ablation of Model Depth

Performance comparison between models with different layer numbers is listed in Table 6. Most of competing models dropped on performance when increasing the layer number to 16, e.g., Graphormer dropped under 50 on HCPA and UKB datasets. In contrast, *NeuroPath* is more stable than others and has the best average ranking.

## 4.7 Ablation of Graph Building

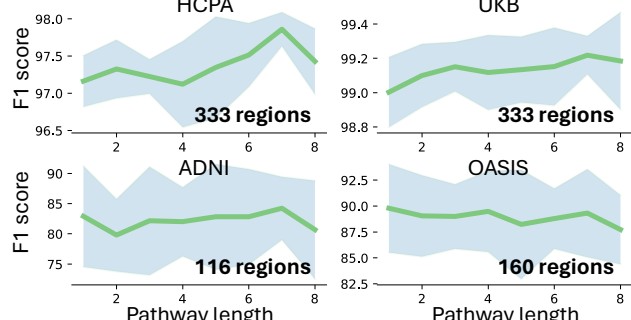

Figure 4: Ablation study of various lengths of the neural pathway that is visible to *NeuroPath*. Static BOLD is set as node attributes in this experiment. The blue shade is the range of error bars and the green lines are average F1 scores.

Table 6: F1 scores by models with different layer numbers on four datasets. '**Bold**' and 'underline' denote the first and the second place of the average rank, respectively.

| | HCPA | | | Rank | UKB | | | Rank |
|---|---|---|---|---|---|---|---|---|
| Layer # | 4 | 8 | 16 | | 4 | 8 | 16 | |
| BNT | 91.81 | 93.41 | 93.28 | 3.67 | 88.63 | 96.32 | 97.45 | 3.00 |
| BolT | 97.01 | 97.81 | 88.23 | 2.33 | 81.36 | 89.20 | 89.84 | 4.00 |
| Graphormer | 64.08 | 47.01 | 50.84 | 5.00 | 43.42 | 43.44 | 59.46 | 5.00 |
| NAGphormer | 96.89 | 97.26 | 97.22 | 2.33 | 99.24 | 98.95 | 99.20 | 2.00 |
| *NeuroPath* | 97.76 | 97.72 | 96.60 | **1.67** | 99.59 | 99.61 | 99.44 | **1.00** |
| | ADNI | | | Rank | OASIS | | | Rank |
| BNT | 76.39 | 75.91 | 77.28 | 3.67 | 85.32 | 85.96 | 85.21 | 3.33 |
| BolT | 75.93 | 78.67 | 78.23 | 2.67 | 85.30 | 84.55 | 85.55 | 3.67 |
| Graphormer | 78.58 | 74.12 | 74.12 | 4.00 | 84.45 | 83.87 | 83.87 | 5.00 |
| NAGphormer | 75.86 | 77.15 | 78.44 | 3.00 | 86.05 | 86.49 | 85.78 | 1.67 |
| *NeuroPath* | 78.93 | 78.42 | 78.32 | **1.67** | 86.16 | 86.77 | 85.78 | **1.00** |

Performance comparison between models with different edge threshold to construct brain network is listed in Table 7. Similarly, existing methods dropped on performance when graph is too dense or too sparse as the threshold of Pearson coefficient changed, which is an empirical hyperparameter. *NeuroPath* can keep the best average ranking in this case.

## 4.8 Pattern of Neural Pathway Contributing to Prediction

As we introduced in Sec. 3.2, *NeuroPath* is in fact weighting neural pathways to obtain the feature representation of the brain network. Therefore, the neural pathways can be sorted by their weights

produced by our model. As shown in Fig. 5, we visualized three pathways corresponding to the same three functionally connected node pairs (blue nodes in Fig. 5), where links that have the same color are members of the same pathway. To show the neural pathway pattern of diseased brain topology, the three functionally connected node pairs are selected from FC links that exhibit significant difference ($p \leq 0.05$) between AD and CN subjects from the OASIS dataset in a $t$-test. We exclude the influence of inter-subject variations of FC mentioned in Sec. 2.1 by narrowing FC links in the $t$-test solely between regions of the subcortical, entorhinal cortex, occipital lobe, and parietal lobe, where those brain structures are associated with the progression of AD [60, 11]. In comparison of pathway visualization in Fig. 5, it is obvious that the AD subject demands a longer SC detour to support the same direct FC link than CN subjects which only need shorter pathways (within two hops). This observation suggests a diseased human connectome might need a longer detour to support normal brain function since the brain network could rewire neurological fibers from other normal regions to fetch up lesion regions [8, 1]. Therefore, this visualization is neuroscience evidence of the interpretability of *NeuroPath*.

## 4.9 Computational Costs

The computational costs can be indicated by the practical running time and the learnable parameter amount as listed in Table 8. Although Graphormer and NAGphormer are two models with lower parameter numbers than *NeuroPath*, they have slower training and testing than *NeuroPath* with pre-processing leading to more computing time. This demonstrates the efficiency of *NeuroPath*.

Table 7: F1 scores by models with different FC thresholds to build brain networks. '**Bold**' and 'underline' denote the first and the second place of the average rank, respectively.

| | HCPA | | | Rank | UKB | | | Rank |
|---|---|---|---|---|---|---|---|---|
| FC threshold | 0.3 | 0.5 | 0.7 | | 0.3 | 0.5 | 0.7 | |
| BNT | 95.73 | 92.57 | 84.51 | 4.00 | 76.41 | 98.64 | 94.46 | 4.33 |
| BolT | 87.02 | 95.78 | 94.68 | 3.00 | 86.98 | 99.29 | 87.04 | 3.67 |
| Graphormer | 90.41 | 53.05 | 88.43 | 4.33 | 97.76 | 86.54 | 96.73 | 3.67 |
| NAGphormer | 96.08 | 94.76 | 96.85 | 2.33 | 97.80 | 99.22 | 98.78 | 2.33 |
| *NeuroPath* | 97.57 | 95.09 | 97.32 | **1.33** | 99.27 | 99.59 | 99.15 | **1.00** |
| | ADNI | | | Rank | OASIS | | | Rank |
| BNT | 77.74 | 80.16 | 77.92 | **1.67** | 85.14 | 85.32 | 86.05 | 3.67 |
| BolT | 74.33 | 76.68 | 76.53 | 4.00 | 84.98 | 84.91 | 84.67 | 4.67 |
| Graphormer | 75.82 | 77.78 | 75.17 | 3.33 | 86.23 | 85.44 | 87.15 | 2.00 |
| NAGphormer | 72.55 | 75.40 | 77.29 | 4.67 | 86.32 | 83.87 | 85.78 | 3.67 |
| *NeuroPath* | 78.36 | 77.35 | 79.49 | **1.67** | 86.59 | 87.02 | 86.13 | **1.33** |

## 5 Conclusion

In this work, we propose *NeuroPath*, a graph transformer to model the physical neural pathway by our novel multivariate SC-FC coupling mechanism and learn the relationship between neural pathways and brain functions. The framework of *NeuroPath* driven by the neuroscience insight can effectively produce SC-FC coupled feature representation of multi-hop neural pathways from the twin branch design. Compared to SOTA brain models and graph transformers on large-scale datasets including HCP and UKB under various data settings, our experiments have not only demonstrated the superior performance of *NeuroPath* in neural activity classification and cognitive disordering diagnosis but also provided great interpretability. Planting the proposed multivariate SC-FC coupling into the design of *NeuroPath* enables it to be not only applicable for zero-shot learning on unseen datasets but also to a better performance than general SOTA models on resting/tasking classification and AD diagnosis. By visualizing the top-1 neural pathway contributing to the prediction by *NeuroPath*, more interpretability is brought to our performance on AD diagnosis, and the observation of visualization agrees with the hypothesis that it needs a longer structural detour to support a functional but diseased human connectome. Modeling neural pathways shows us a clue of fundamental neuroscience models to decipher the relationship between brain structural and functional topology.

Table 8: Computational complexity in our experiments, where computing time is the average on UKB dataset with unit the millisecond per graph data.

| | Param # | Preproc. | Train | Test |
|---|---|---|---|---|
| BrainGNN | 7.30M | - | 7.24 | 2.61 |
| BNT | 1.57M | - | 1.82 | 0.64 |
| BolT | 1.58M | - | 3.83 | 1.83 |
| Graphormer | 0.30M | 270 | 2.79 | 0.90 |
| NAGphormer | 0.26M | 40 | 3.92 | 1.85 |
| *NeuroPath* | 0.69M | - | 1.61 | 0.67 |

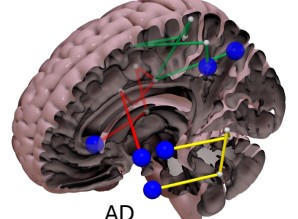 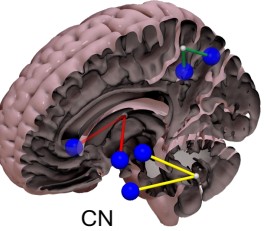

AD CN

Figure 5: The visualization of the top-1 neural pathway that corresponds to a significant FC link contributing to the prediction by *NeuroPath* on OASIS dataset.

# 6   Acknowledgments

Thanks to the Foundation of Hope and NIH grant AG068399.

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

# A  Appendix

## A.1  Accessibility

All data can be accessible via internet (HCPA[2], UKB[3], ADNI[4], OASIS[5]). The licenses to obtain those data can also be accessed on the websites. The preprocessing algorithm is introduced in the next section. The codes and data split settings can be acquired via this code repository[6].

## A.2  Data Preprocessing

The neuroimage processing consists of the following major steps:

- We segment the T1-weighted image into white matter, gray matter, and cerebral spinal fluid using FSL software [27]. On top of the tissue segmentation in Fig. 6, we parcellate the cortical surface of fMRI into cortical regions according to the different atlas as a regional signal of timeseries in Fig. 6, where FC, in the end, is the Pearson correlation coefficient between regional timeseries. Additionally, we convert each DWI scan to diffusion tensor images (DTI) [54] in this step.

- We apply surface seed-based probabilistic fiber tractography [18] using the DTI data, thus producing an anatomical connectivity matrix (SC in Fig. 6). Note that the weight of the anatomical connectivity is defined by the number of fibers linking two brain regions normalized by the total number of fibers in the whole brain.

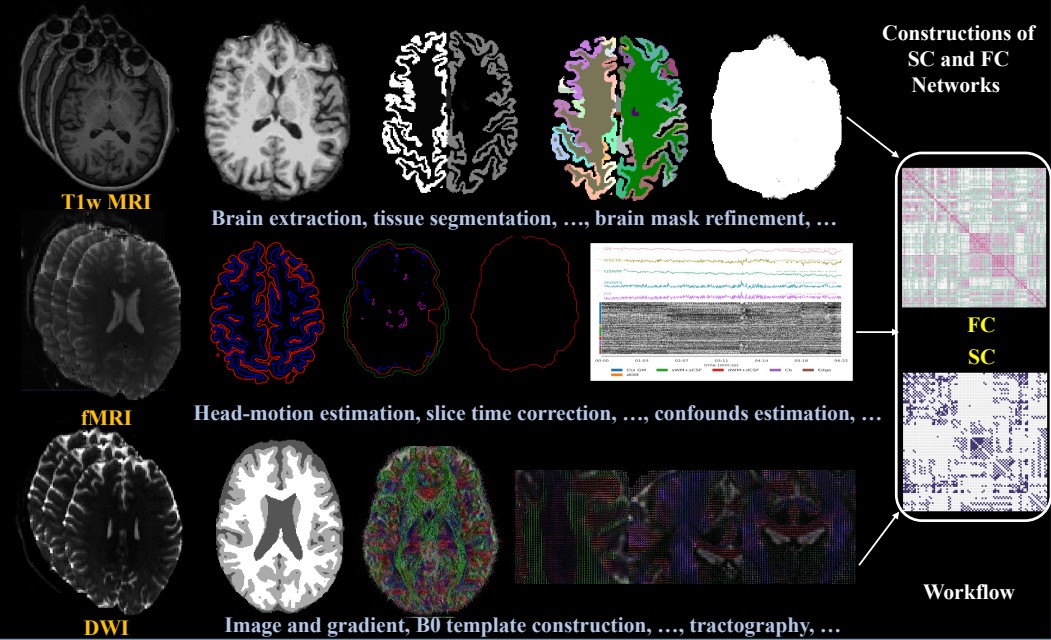

Figure 6: General workflows for processing T1-weighted image (T1w MRI), functional MRI (fMRI), and diffusion-weighted image (DWI). The output is shown at the right, including two brain networks of FC and SC.

---

[2]https://www.humanconnectome.org/

[3]https://www.ukbiobank.ac.uk/

[4]https://adni.loni.usc.edu/

[5]https://sites.wustl.edu/oasisbrains/

[6]https://github.com/chrisa142857/neuro_detour

## A.3  Data Profiles

Data profiles of four datasets are listed in this section. As listed in Table 9, the graph number of static data could be larger than the subject number in the corresponding dataset introduced in the main text since few subjects have multiple runs/sessions in datasets, where the time gap between runs/sessions is large so they are preprocessed as independent data. Dynamic data is split from the full scan of one subject into 100-length slices causing more graphs and a different connectivity.

One major phenotype of brain networks is the age of the subject as shown in Fig. 7. Accordingly, our experiments are setup with a wide range of ages.

Table 9: Data profiles, where $|G|$ denotes the number of graphs, $|C|$ denotes the number of classes, and $avg(D)$ denotes the average degree of brain networks.

|  | HCPA | | UKB | | ADNI | | OASIS | |
|---|---|---|---|---|---|---|---|---|
|  | static | dynamic | static | dynamic | static | dynamic | static | dynamic |
| $|G|$ | 4,863 | 18,306 | 5,890 | 22,600 | 138 | 294 | 402 | 1,678 |
| $|C|$ | 4 | 4 | 2 | 2 | 2 | 2 | 2 | 2 |
| $avg(D)$ | 6.53 | 13.52 | 12.85 | 36.07 | 44.44 | 43.89 | 56.47 | 59.36 |

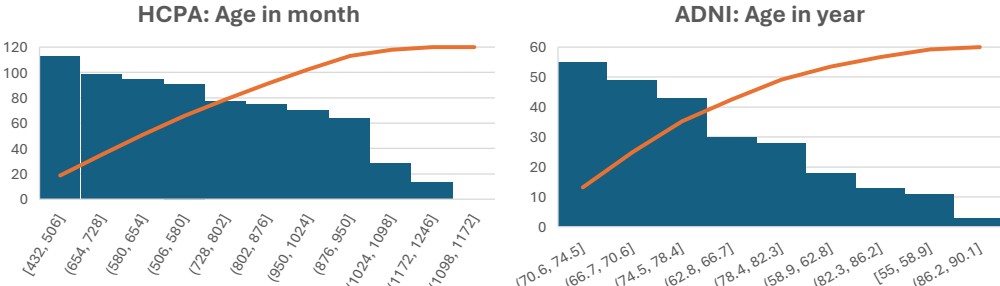

Figure 7: The age distribution of HCPA and ADNI datasets, where blue bars are histograms of age, and orange curves denote the data percentage accumulation.

## A.4  Detailed Steps of Paired *t*-test

A paired *t*-test is run for the topological detour degree found by the depth-first algorithm with a radius of 6 for every parcel per subject between resting and tasking groups, where there are 716 subjects in total from the HCPA dataset. The vector of a parcel from all subjects in the same group is the input to *t*-test algorithm, where subjects of two groups are strictly paired. In comparison, the degree of FC is calculated by the total number of FC links for every parcel and tested in the same way using *t*-test algorithm. After *t*-test, significant parcels are obtained by filtering the *p*-value smaller than 0.05. Gordon parcellation has a fine partition ($n$=330) of the brain and has 12 pre-defined sub-networks relevant to brain functions. The overlap ratio is calculated by $\frac{N_{sign_i}}{N_{net_i}}$, where $N_{sign_i}$ is the number of parcels of $i$-th community have $p \leq 0.05$ and $N_{net_i}$ is the total number of parcels in $i$-th sub-network.

## A.5  Paired *t*-test on diverse populations by gender on HCPA dataset

As shown in 8, we run the same *t*-test as in Fig.2 in the main text for the degree of our topological detour across diverse populations by gender. By comparing the detour degree with the FC degree, we can draw the same conclusion as in Sec 2.1. It is worth noting that the male group shows a lower detour degree, i.e., fewer detour pathways, than the female group in the HCPA dataset.

## A.6  Proof of Fact

**Fact 3.1** *The top pathway representations are obtained by* $\arg\max_{j,h} \left( \frac{1}{h} \sum_{j \in \mathbf{p}} \mathbf{S}_{ij} \bar{\gamma}_h \bar{\mathbf{W}}_{i \sim j} \right)$, *where* $\mathbf{S}$ *denotes the softmax of self-attention,* $\mathbf{p} \subset \mathbf{P}_i^H$ *is a set of node index of a path and* $\mathbf{P}_i^H$ *is the node collection of neural pathways within* $H$-*hop starting at* $i$-*th node.*

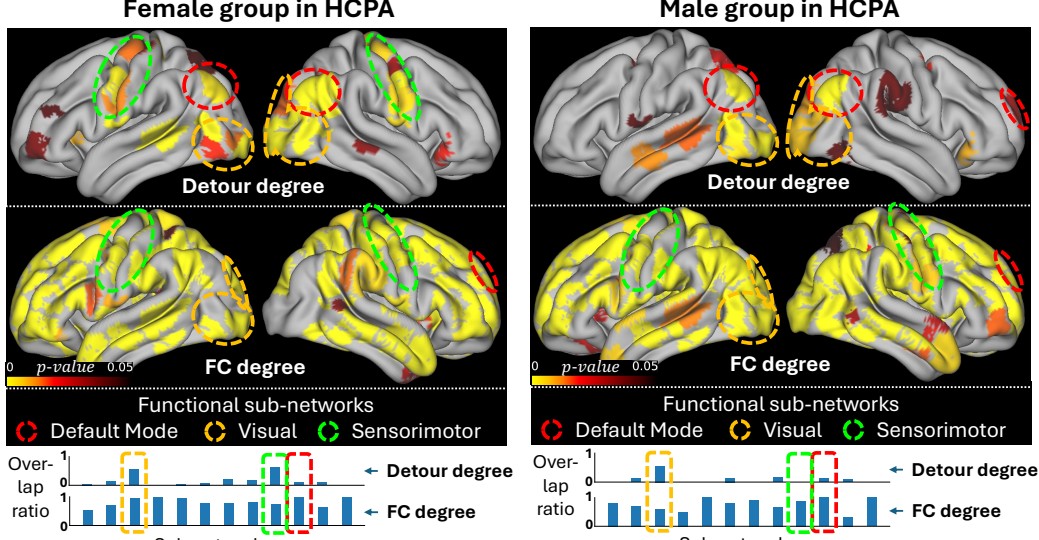

Figure 8: The same $t$-test as Fig.2 in the main text runs on two diverse populations by gender among HCPA dataset.

*Proof.* The first is to expand Eq. 1 by using definitions mentioned in Section 3.

$$
\begin{aligned}
f_{TD}(\mathbf{X}) &= \texttt{Concat}\left(\bar{\mathbf{X}}_1, \ldots, \bar{\mathbf{X}}_H\right) \bar{\mathbf{W}} \\
&= \sum_{h=1}^{H} \bar{\mathbf{X}}_h \bar{\mathbf{W}}_{i \sim j}, i := (h-1) * C, j := h * C,
\end{aligned}
\tag{4}
$$

where $\bar{\mathbf{W}}_{i \sim j}$ is a slice of weights $\bar{\mathbf{W}}$ with $C$ entries, and we have

$$
\bar{\mathbf{X}}_h = \texttt{Softmax}\left((\mathbf{X}\bar{\boldsymbol{\alpha}}_h)(\mathbf{X}\bar{\boldsymbol{\beta}}_h)^T + f_{mask}(\mathbf{D}^h)\right)\left(\mathbf{X}\bar{\boldsymbol{\gamma}}_h\right).
\tag{5}
$$

In plain words, $\texttt{Softmax}$ in the above equation is the standard softmax operation that only involves $h$-hop connected columns and applies on every row, then it produces a self-attention matrix $\mathbf{S} \in \mathbb{R}^{N \times N}$, where

$$
\mathbf{S}_{ij} = \begin{cases} \texttt{Softmax}\left(x_i\right)_j & \text{if } j \in \text{argtrue}\mathbf{D}_{i,\cdot}^h, \\ 0 & \text{otherwise.} \end{cases}
\tag{6}
$$

where $x$ denotes the term inside $\texttt{Softmax}$ in Eq. 5. Then

$$
f_{TD}(\mathbf{X})_i = \sum_{h=1}^{H} \sum_{j \in \text{argtrue}\mathbf{D}_{i,\cdot}^h} \mathbf{S}_{ij} \mathbf{X}_j \bar{\boldsymbol{\gamma}}_h \bar{\mathbf{W}}_{i \sim j}.
\tag{7}
$$

Herein, this proof is transformed to representing node sets of all pathways by $\{\text{argtrue}\mathbf{D}_{i,\cdot}^h\}_{h=1,\ldots,H}$, i.e., the following Lemma

**Lemma A.1.** *The node collection of half of the neural pathways within $H$-hop started at the $i$-th node can be sorted as $\mathbf{P}_i^H \in \{argtrue\mathbf{D}_{i,\cdot}^h\}_{h=1,\ldots,H}$, where 'collection' denotes a set allowing repeated members.*

To prove this Lemma, there is an easy fact that can be seen. If $\forall i$ the degree $\mathbf{D}_i > 0$, then

$$
(\hat{\mathbf{A}}^{\mathbf{S}})_{ij}^h > 0 \implies (\hat{\mathbf{A}}^{\mathbf{S}})_{ij}^{h+2} > 0, \forall h > 0
\tag{8}
$$

This is true since $\forall 1 \leq i \leq N, (\hat{\mathbf{A}}^{\mathbf{S}})_{ii}^2 > 0$. Then, given that $\mathbf{D}_{i,\cdot}^h := \left((\hat{\mathbf{A}}^{\mathbf{S}})^h > 0\right) \cdot \left(\hat{\mathbf{A}}^{\mathbf{F}}\right)$ and $\hat{\mathbf{A}}^{\mathbf{F}}$ remains unchanged with various $h$, we can have the same deduction on detour adjacency matrix

$$
\mathbf{D}_{ij}^h > 0 \implies \mathbf{D}_{ij}^{h+2} > 0, \forall h > 0
\tag{9}
$$

Given Eq. 9, $\forall \mathbf{D}_{ij}^h > 0$, the $j$-th node will present accumulated on every other hop adjacency matrix, i.e., $\hat{\mathbf{A}}_{ij}^{\mathbf{D}^k} > 0$ if $k = h + 2t$ and $t \in \mathbb{Z}^+$. Thus, the node index $j$ as a member of multiple/single pathways with at least a length of $h$ appears $(H - h)//2 + 1$ times.

Assuming a path sub-graph starts at $i$-th node ends at $h$-hop and has the node index collection $\{i+1, i+2, \cdots, i+t, \cdots, i+h\}$ regardless of the starting node, then in this case,

$$\mathbf{P}_i^H = \{[i+t] * ((h-t)//2 + 1)\}_{t=1,\cdots,h}, \tag{10}$$

where $[\cdot] * x$ means repeating elements $x$ times in a collection. Since the entire node index collection of one path is $\{[i+t] * (h-t+1)\}_{t=1,\cdots,h}$, $\mathbf{P}_i^H$ can be sorted as exactly the half of all simple paths in this path sub-graph if it contains even number of nodes, or half + 1 if the number is odd. Consequently, we can make a deduction from one path to all non-overlap paths started at $i$-th node, where the definition of the neural pathway in our work is the non-overlap paths.

Taking Eq. 7 and Lemma A.1 together, $f_{TD}(\mathbf{X})_i$ weights all nodes of half of the neural pathways by sorting results $\mathbf{P}_i^H$. By assigning each node with the learnable weights $\mathbf{S}_{ij} \bar{\gamma}_h \bar{\mathbf{W}}_{i \sim j}$ since $\mathbf{S}_{ij}$ is a scalar that can be moved aside to matrices, each path weights is thus represented by this weight consists of learnable parameters and can be extracted to explain the contribution by neural pathways to any downstream application. This can finish the proof. $\qquad \square$

## A.7 Hardwares

Our experiments are run on a local platform that has dual Intel(R) Xeon(R) Gold 6448Y CPUs and four NVIDIA RTX 6000 Ada GPUs. All SOTA models in our experiments used the default hyperparameters except for Graphormer which is implemented by us. Detailed settings can be found in the released codes.

## A.8 Limitations and Discussions

The limitations of *NeuroPath* are two folds. *First*, the top path representation is limited on the half number of neural pathways in theory. This limitation is amplified in dynamic data that has a higher degree of connections leading to more pathways, where the performance is relatively lower than in static data as listed in the tables of our experiments. Although our detour adjacency matrix avoids finding all simple paths in terms of computational complexity, modeling a brain network with all neural pathways could make *NeuroPath* more fundamental. *Second* is the limited size of disease datasets used in the experiments.

## A.9 Societal Impact

Our major contribution to the machine learning field is a novel graph transformer framework that has sufficient expressive power for the neural pathway of human connectome with a much lower computational complexity than previous path neural networks. This allows us to develop a new deep model with a novel multivariate SC-FC coupling mechanism. Furthermore, we have provided zero-shot learning experiments to demonstrate the potential of our model to be a fundamental neuroscience model. From the application perspective, the new deep model for uncovering the physical neural pathway supporting the *in-vivo* functional connection has great potential to establish a new underpinning of the relationship between brain structure and function topology.

