# OpenReview forum: "$\textit{NeuroPath}$: A Neural Pathway Transformer for Joining the Dots of Human Connectomes"
_NeurIPS.cc/2024/Conference — NeurIPS 2024 poster_

### Official Review · Reviewer_g82y · 2024-06-21

**Soundness:** 3
**Presentation:** 2
**Contribution:** 2
**Rating:** 6
**Confidence:** 3

**Summary:**

In this work, the authors propose NeuroPath, a graph transformer acting on structural (SC) and functional (FC) connectivity matrices. NeuroPath aims to learn the relationship between these pathways and brain functions. The framework uses a twin-branch design to generate coupled features of multi-hop neural pathways.

Experimental results on large datasets such as HCP and UKB show that NeuroPath outperforms state-of-the-art (SOTA) brain models and graph transformers in tasks like neural activity classification and cognitive disorder diagnosis. It offers superior performance in resting/tasking classification and Alzheimer's Disease (AD) diagnosis, showing promise for zero-shot learning on unseen datasets.

The model's interpretability is enhanced by visualizing the top neural pathways contributing to its predictions.

**Strengths:**

-	Comprehensive Integration of Connectivity Matrices: incorporates multi-hop structural and functional connectivity matrices, enabling to investigate the interplay between these two types of neural interactions. This approach addresses a significant gap in understanding brain function, as the relationship between structural and functional connectivity remains a complex and open research problem in neuroscience.
-	Extensive Experimental Evaluation: NeuroPath's effectiveness is demonstrated through extensive experiments on large-scale datasets, including thorough comparisons with state-of-the-art (SOTA) and traditional models. The results consistently show that NeuroPath achieves superior performance, highlighting its potential to advance the field of neural activity classification and cognitive disorder diagnosis.

**Weaknesses:**

-	Thresholding Ambiguity: Structural Connectivity (SC) and Functional Connectivity (FC) matrices are significantly influenced by the thresholding applied to filter their entries. The paper lacks clarity on how this thresholding process was implemented and whether a consistent threshold was applied across all models tested. This omission raises concerns about the comparability and validity of the experimental results.
-	Lack of Clarity on Theoretical Foundations: Fact 3.1 is insufficiently explained, and its proof in the Appendix is difficult to follow. The paper does not clearly define the "expressive power" of the model, leaving readers without a solid understanding of this key concept and its implications for the model’s performance. This lack of clarity diminishes the strength of the theoretical contributions and the overall comprehensibility of the work.

**Questions:**

-	Clarification on Thresholding Methodology: What specific thresholding techniques were used to derive the structural and functional connectivity matrices? Could you provide details on the criteria and process for selecting these thresholds?
-	Consistency Across Models: Were identical thresholds applied to the connectivity matrices across all models tested in your experiments? If different thresholds were used, please elaborate on the reasoning behind this choice and the potential impact on the results.
-	Impact of Threshold Variation: Is there a possibility that adjusting the thresholds applied to the connectivity matrices could result in other models outperforming NeuroPath in predictive performance? Have you conducted any experiments to explore the sensitivity of the results to different threshold levels?
-	Definition of Model's Expressive Power: How do the authors specifically define the "expressive power" of the model in the context of your study? Could you provide a clearer explanation of this concept and how it relates to the model’s ability to capture and represent complex neural interactions?
-	Line 129: "Transform"-> "Transformer"
-	Line 163: "Production"-> "Product"

**Limitations:**

Yes, the authors included short sections in the appendix covering the Limitations and societal impact of their work.

---

> ### Author Rebuttal · Authors · 2024-08-06
>
> **W1, Q1, and Q2:** Consistent thresholds are applied across all experiments and t-tests in our manuscript for all models and baselines. Thresholds for FC and SC are consistently set as 0.5 and 0.1, respectively, for all models and datasets, where SC is normalized to [0, 1] before thresholding.
>
> **Q3:** To show the threshold variation, we adjust the threshold to 0.3 and 0.7 leading to more and fewer edges in the FC graphs for all four datasets, respectively. As listed below, some baselines could be very sensitive to thresholding. Such as BNT, BolT, and Graphormer show significant drops when the FC threshold changes. This pattern is clearly shown in the second row of Fig.S3 in our newly uploaded PDF. In contrast, NeuroPath always shows the best average rank of performance.
>
> |Model       |HCPA |     |     |Rank|UKB  |     |     |Rank|
> |------------|-----|-----|-----|----|-----|-----|-----|----|
> |FC threshold|0.3  |0.5 |0.7  |    |0.3  |0.5|0.7  |    |
> |BNT         |95.73|92.57|84.51|4.00|76.41|98.64|94.46|4.33|
> |BolT        |87.02|95.78|94.68|3.00|86.98|99.29|87.04|3.67|
> |Graphormer  |90.41|53.05|88.43|4.33|97.76|86.54|96.73|3.67|
> |NAGphormer  |96.08|94.76|96.85|2.33|97.80|99.22|98.78|2.33|
> |NeuroPath   |97.57|95.09|97.32|**1.33**|99.27|99.59|99.15|**1.00**|
>
> |       |ADNI |     |     |Rank|OASIS  |     |     |Rank|
> |------------|-----|-----|-----|----|-----|-----|-----|----|
> |BNT         |77.74|80.16|77.92|1.67|85.14|85.32|86.05|3.67|
> |BolT        |74.33|76.68|76.53|4.00|84.98|84.91|84.67|4.67|
> |Graphormer  |75.82|77.78|75.17|3.33|86.23|85.44|87.15|2.00|
> |NAGphormer  |72.55|75.40|77.29|4.67|86.32|83.87|85.78|3.67|
> |NeuroPath   |78.36|77.35|79.49|**1.67**|86.59|87.02|86.13|**1.33**|
>
>
> **W2, and Q4:** Thank you for raising the important concern of using an unexplained term in our theoretical analysis. In Sec 3.2, we explored the theoretical foundation of our NeuroPath by using expressive power which appears not the best term. The core of our theoretical analysis is how many neural pathways are allowed to be represented by NeuroPath as we described in the paragraph next to Fact 3.1. However, expressive power refers to the ability of a graph model to distinguish the isomorphism of graph (sub)structure. This technical term is popularly used to prove the expressiveness of a GNN [1]. Since we focus on neural pathways modeling instead of isomorphism of (sub)structure, expressive power does not  exactly fit to frame our theory for modeling complex neural interactions. Instead, we will revise Fact 3.1 by replacing the expressive power with the number of the path substructure that can be modeled by NeuroPath. Based on this, in our proof of Fact 3.1, such capacity of modeling half paths of a graph can be easily formulated as Lemma A.1 after expanding Eq (1) as Eq (7). Then, it is easy to follow the proof of Lemma A.1 after we get Eq (9).
>
> [1] 10.1007/978-981-16-6054-2_5

---

> > ### Comment · Reviewer_g82y · 2024-08-10
> > **Thanks**
> >
> > Thank you to the authors for addressing my questions and providing clear responses. However, I will be keeping my original score.

---

> > > ### Author Response · Authors · 2024-08-10
> > > **Response to Reviewer g82y**
> > >
> > > Dear, Reviewer,
> > >
> > >     We appreciate that.
> > >
> > >     Have a nice weekend.
> > >
> > > Best,
> > >
> > > Authors

---

### Official Review · Reviewer_6veV · 2024-07-12

**Soundness:** 3
**Presentation:** 3
**Contribution:** 2
**Rating:** 6
**Confidence:** 3

**Summary:**

Authors propose NeuroPath, a transformer-inspired model that leverages a multi-head self-attention mechanism to capture multi-modal feature representations from SC (structural connectivity) and FC (functional connectivity) graphs. The model is evaluated on well-known large-scale public datasets OASIS, ADNI, UK Biobank, and HCPA. One of the main novelties of the paper is the use of coupled SC and FC graphs.

**Strengths:**

This paper introduces the concept of *topological detour* to characterize how functional connectivity (FC) is supported by neural pathways in structural connectivity (SC). This approach goes beyond the traditional univariate coupling, allowing the model to capture complex, multi-hop pathways that support functional connectivity. The proposed model architecture uses a *twin-branch approach* to accommodate the SC-FC pairing.

**Weaknesses:**

While the authors do include a table with the number of learnable parameters, it would be nice to include explicit information about the training procedures and the necessary time to train the model.

**Questions:**

Would you have more information on the demographics of the datasets included in the paper?

**Limitations:**

While the model aims to handle inter-subject variations, it's unclear how well it accounts for individual differences in brain structure and function across diverse populations. It would be nice if the authors could include a discussion about that.

---

> ### Author Rebuttal · Authors · 2024-08-06
>
> ## Weaknesses
> The actual computing time of existing brain models, transformers, and our NeuroPath per graph is shown below.  According to this table, the computational time to train the model of every experiment can be calculated along with the data number in Table 5 in our manuscript.
>
> |          |Param #|Pre-proc time / graph|Train / graph|Test / graph|
> |----------|-------|---------------------|-------------|------------|
> |BrainGNN  |7.30M  |\-                   |7.24ms       |2.61ms      |
> |BNT       |1.57M  |\-                   |1.82ms       |0.64ms      |
> |BolT      |1.58M  |\-                   |3.83ms       |1.83ms      |
> |Graphormer|0.30M  |270ms                |2.79ms       |0.90ms      |
> |NAGphormer|0.26M  |40ms                 |3.92ms       |1.85ms      |
> |NeuroPath |0.69M  |\-                   |1.61ms       |0.67ms      |
>
> ## Questions
>
> Yes, we have. Distributions of the subject age are shown **in Fig.S1** in our newly uploaded PDF file.
>
> ## Limitations
>
> **In Fig.S2**, We run the same $t$-test as in Fig.2 in the main text for the degree of our topological detour across diverse populations by gender. By comparing the detour degree with the FC degree, we can draw the same conclusion as in Sec 2.1. It is worth noting that the male group shows a lower detour degree, i.e., fewer detour pathways, than the female group in the HCPA dataset.

---

### Official Review · Reviewer_Qyot · 2024-07-13

**Soundness:** 3
**Presentation:** 3
**Contribution:** 3
**Rating:** 6
**Confidence:** 3

**Summary:**

This paper introduces a novel way of predicting brain diseases with the help of structural and functional brain connectivity. It couples structural as well as functional connectivity from human neuroimaging studies. It comprehensively studies the performance of the new method on a large variety of datasets. In all, the study is quite comprehensive in terms of experiments.

**Strengths:**

There are a lot of experiments on a variety of different datasets and hence this makes it a very robust method for mentioning structural and functional connectivity. Most results are also statistically significant with p < 0.05. the model structure is also well mathematically described.

**Weaknesses:**

Some mathematical details are not very understandable Framework of twin branch, FC-MHSA etc. Equations on line 173 are not easy to understand. The results of the ablation studies are not presented in a very comprehensive manner.

**Questions:**

The ablation studies would be better described with visual information and other properties.

**Limitations:**

See weaknesses

---

> ### Author Rebuttal · Authors · 2024-08-06
>
> ## Weaknesses
>
> Equations on line 173 can be rewritten more clearly: a set of learnable parameters $\{ \bar{\mathbf{W}}, \hat{\mathbf{W}} \in \mathbb{R}^{(HC)\times C} \}$ and $\boldsymbol{\bar\alpha}_h, \boldsymbol{\bar\beta}_h, \boldsymbol{\bar\gamma}_h, \boldsymbol{\hat\alpha}_h, \boldsymbol{\hat\beta}_h, \boldsymbol{\hat\gamma}_h \in \mathbb{R}^{C\times C}$ where $h=1,\dots,H$.
>
> ## Questions
>
> We run more ablation studies on other properties by scaling the model size and varying the threshold of FC graph construction. They are visualized by line plots for a better description as shown in Fig.S3 in our newly uploaded PDF file.

---

> > ### Comment · Reviewer_Qyot · 2024-08-11
> > **Thanks**
> >
> > Dear authors,
> >
> > Thank you very much for your comments. I, however, would like to stick to my original scores.

---

> > > ### Author Response · Authors · 2024-08-11
> > >
> > > Dear, Reviewer,
> > >
> > > We appreciate that.
> > >
> > > Have a nice weekend.
> > >
> > > Best,
> > > Authors

---

### Official Review · Reviewer_4uWQ · 2024-07-13

**Soundness:** 3
**Presentation:** 3
**Contribution:** 2
**Rating:** 5
**Confidence:** 4

**Summary:**

This paper introduces a transformer model that integrate both structural connectivity (SC) and functional connectivity (FC). It formulates a graph representation learning framework to extract feature from the brain connectome data. The model has two branches to encode SC and FC data separately, and later training to align the two modalities with consistency constrain loss. The learned representations could be further applied into multiple downstream tasks, including neural activity classification, cognition disordering diagnosis, etc. It also demonstrates its performance on zero-shot learning.

**Strengths:**

**Motivation**
1. The paper is well motivated to integrate both functional connectivity and structural connectivity in brain connectome data to improve the performance in the downstream tasks.

**Method**
1. This model has focused to develop an efficient model with half of the expressive power of PathNN, and also provided with a theoretical proof.
2. The pattern of neural pathway to help interpret the model's prediction adds more strengths to the proposed method.

**Evaluation**
1. This work did extensive evaluation on multiple benchmarks and multiple baselines, and performed ablation studies to demonstrate the effectiveness of the having two branches of NeuroPath model.
2. The demonstration of model's capability on zero-shot learning is interesting.

**Weaknesses:**

**Novelty**
1. The model has limited novelty compared to existing SOTA models using transformer, and graph transformer model on brain connectome data.

**Peformance**
1. The model also has limited improvement on the model performance. The accuracy is on par or worse than SOTA baseline in multiple downstream tasks.
2. Though the authors highlight the efficiency of model designs, while the proposed model's size (0.69M) is larger than some baselines including Graphormer (0.3M), NAGphormer (0.26M) which achieved comparable performance.

**Questions:**

1. How is the temporal information and complex dynamics of functional connectivity modeled in the framework? What is the limitation for the window size for the dynamics to be captured?
2. What is the scalability of the proposed model?
3. Discuss choices for hyperparameters?
4. Discuss major differences in the developed method with compared baselines?

**Limitations:**

The paper does not have potential negative societal impact. The limitation of modeling dynamical data and limited data size has been thoroughly discussed. Justifications for 9, 11, 13, 14, 15 are missing in checklists.

---

> ### Author Rebuttal · Authors · 2024-08-06
>
> ## Novelty
> 1. Although there are previous works utilized transformer and graph transformer, our *NeuroPath* is the first model to uncover the SC-FC coupling mechanism between (structural) neural pathways and (functional) neural activities under a new design of MHSA framework that can represent the path of graph without any pre-processing.
>
>  - As highlighted by the other reviewers, our work introduced a novel comprehensive integration of connectivity matrices coined as **topological detour**, a novel graph substructure showing a significant contribution to neuroscience research of structure-function coupling. Our *NeuroPath* method is the first model that can learn the relationship between such detour pathways and brain activity.
>
>  - On the other hand, existing frameworks of modeling the path of a graph mainly focus on introducing high-order features [1] or grouping nodes of a path [2]. Our *NeuroPath*, in contrast, does not require any pre-processing to obtain features or search paths in advance. This makes the pathway modeling implementable since the brain connectome graph is so dense that finding all paths is highly time-consuming as listed in Table below.
>
> |     | PathNN  | Graphormer  | NAGphormer | *NeuroPath* |
> | - | - | - | - | - |
> | Pre-process type         | All simple paths        | Shortest distance | Graph diffusion | None      |
> | Pre-process time / graph | 5.23s ($H=4$), 650s ($H=5$) | 270ms             | 40ms            | \-        |
>
>
> ## Performance
> 1. In our manuscript, we aim to show comprehensive results that contain all possible training scenarios of brain connectome data to test both accuracy and robustness. This follows the previous benchmarking work [3]. For the sake of clarity, we calculate the average rank of performance to show a comprehensive performance rank of *NeuroPath* and 8 baselines as below, where *NeuroPath* shows the best average rank on all four datasets. On the other hand, we shown a more practical downstream task in the manuscript, zero-shot learning, where results also show a significant improvement in performance for all datasets, e.g., more than 16% improvement against the second-place method when training on HCPA and testing on UKB. Moreover, by varying the model size and FC threshold as shown in Fig.S3 in our newly uploaded PDF, *NeuroPath* still has the best comprehensive performance.
>
> |  | HCPA | UKB  | ADNI | OASIS |
> | - | - | - | - | - |
> | MLP   | 4.0  | 3.0  | 6.9  | 3.3   |
> | GCN   | 4.5  | 3.75 | 4.4  | 5.3   |
> | BrainGNN   | 7.0  | 7.0  | 4.1  | 4.4   |
> | BNT   | 2.8  | 5.0  | 2.1  | 2.8   |
> | BolT  | 2.5  | 2.25 | 5.8  | 6.5   |
> | Graphormer   | 8.0  | 8.0  | 4.6  | 6.4   |
> | NAGphormer   | 5.3  | 5.3  | 5.8  | 5.0   |
> | *NeuroPath*    | **2.0**  | **1.8**  | **1.6**  | **2.5**   |
>
> 2. The efficiency of our *NeuroPath* is demonstrated by the parameter number and the actual computing time. The table below shows the average processing time on the UKB dataset.
>
> |   | Param # | Pre-proc time / graph | Train / graph | Test / graph |
> | - | - | - | - | - |
> | Graphormer | 0.30M   | 270ms   | 2.79ms        | 0.90ms       |
> | NAGphormer | 0.26M   | 40ms     | 3.92ms        | 1.85ms       |
> | *NeuroPath*  | 0.69M   | \-     | **1.61ms**        | **0.67ms**       |
>
> ## Questions
> **1** We thank for these insightful comments. We are fully aware of the importance of functional dynamics in the network neuroscience field. In our current implementation, we use the time series of the BOLD signal as the node feature to learn temporal information in *NeuroPath*. Although we have not employed sliding window techniques, we specifically evaluate the effect of window size in our deep model. As we showed in the experiment section (ln Sec 4), we have not found statistical significance with respect to window size (from 100 to 500 time points). In this paper, we put the spotlight on the concept of **topological detour** for SC-FC coupling. Incorporating sliding windows is definitely on our radar for future work.
>
> **2** We scale the model size and compare the scalability between *NeuroPath* and the existing graph transformers and brain transformers that have more layers as listed below and in Fig.S3.
>
> |Model     |HCPA |     |     |Rank|UKB  |     |     |Rank|
> |-|-|-|-|-|-|-|-|-|
> |Layer #   |4    |8    |16   |    |4    |8    |16   |    |
> |BNT |91.81|93.41|93.28|3.67|88.63|96.32|97.45|3.00|
> |BolT    |97.01|97.81|88.23|2.33|81.36|89.20|89.84|4.00|
> |Graphormer|64.08|47.01|50.84|5.00|43.42|43.44|59.46|5.00|
> |NAGphormer|96.89|97.26|97.22|2.33|99.24|98.95|99.20|2.00|
> |*NeuroPath* |97.76|97.72|96.60|**1.67**|99.59|99.61|99.44|**1.00**|
>
> | |ADNI |     |     |Rank|OASIS  |     |     |Rank|
> |-|-|-|-|-|-|-|-|-|
> |BNT  |76.39|75.91|77.28|3.67|85.32|85.96|85.21|3.33|
> |BolT |75.93|78.67|78.23|2.67|85.30|84.55|85.55|3.67|
> |Graphormer|78.58|74.12|74.12|4.00|84.45|83.87|83.87|5.00|
> |NAGphormer|75.86|77.15|78.44|3.00|86.05|86.49|85.78|1.67|
> |*NeuroPath* |78.93|78.42|78.32|**1.67**|86.16|86.77|85.78|**1.00**|
>
> **3** There is only one hyperparameter of *NeuroPath* related to the representation of neural pathways, $H$, the hop # of TD-MHSA. We show the best choice of this hyperparameter in Fig.4 in the manuscript, where the best $H$ is decided by brain region number.
>
> **4** In the response of **Novelty**, we discussed the difference between *NeuroPath* and existing general graph transformers. The difference between *NeuroPath* and brain models is mainly the objective of brain connectomes data modeling. As discussed above, we focus on modeling brain activity via neural pathways acquired from structure-function integrated connectivity. In contrast, existing brain models are whether merging [4,5] nodes of a graph to aggregate information of brain subnetworks or fusing [6] dynamic brain time series.
>
> [1] 10.1109/tpami.2022.3154319
>
> [2] 10.48550/arxiv.2306.05955
>
> [3] 10.48550/arxiv.2306.06202
>
> [4] Brain Network Transformer
>
> [5] 10.1016/j.media.2021.102233
>
> [6] 10.1016/j.media.2023.102841

---

> ### Comment · Reviewer_4uWQ · 2024-08-12
>
> Thanks for the authors' for clarifications and additional details on novelty and performance. And the advantages of not involving preprocessing to extract features, and include structure-function coupling. I have increased my score correspondingly.

---

### Author Rebuttal · Authors · 2024-08-06

# Thank for the insightful comments from all reviewers.

## Performance concern

Since we aim to comprehensively test various scenarios of modeling brain activities, we run baselines and our NeuroPath using multiple experimental settings on each dataset. However, we simply list all numbers to show the model performance in the manuscript confusing readers to get a conclusion. We will revise the final version by adding the average rank of every method as one comprehensive performance rank on each dataset.

## Additionally, there are three new experiments and three new figures in the newly uploaded PDF to support our response.

 - Exp1: Model size scalability comparison between baselines and our *NeuroPath* that have more layers than in the manuscript.

 - Exp2: Performance stability comparison between baselines and our *NeuroPath* on all datasets with different FC thresholds.

 - Exp3: The same $t$-test as in Sec 2.1 runs on diverse populations by gender among the HCPA dataset to compare the inter-subject variations between the FC degree and our **detour** degree.

 - FigS1: Demographics of datasets.

 - FigS2: Results of Exp3

 - FigS3: Results of Exp1 and Exp2 are shown in line plots

We will include new results into the final version by adding to Tables 1, 2, and the Appendix.

---

### Decision · Program_Chairs · 2024-09-25

**Decision:**

Accept (poster)

**Comment:**

This submission generated interest from the reviewers. They appreciated the solid and extensive empirical evaluation of the contributed model, that couples structural and functional connectivity. In addition, the reviewers saw benefits in the zero-shot capabilities, and decreased preprocessing needs. The limited improvement in model performance was noted.